# HDL Accessory Proteins in Parkinson’s Disease—Focusing on Clusterin (Apolipoprotein J) in Regard to Its Involvement in Pathology and Diagnostics—A Review

**DOI:** 10.3390/antiox11030524

**Published:** 2022-03-09

**Authors:** Izabela Berdowska, Małgorzata Matusiewicz, Małgorzata Krzystek-Korpacka

**Affiliations:** Department of Medical Biochemistry, Wroclaw Medical University, Chałubińskiego 10, 50-368 Wroclaw, Poland; malgorzata.krzystek-korpacka@umw.edu.pl

**Keywords:** Parkinson’s disease, apolipoprotein J, ApoJ, clusterin, paraoxonase, PON1, ApoA1, ApoD, ApoE, neurodegenerative disorders

## Abstract

Parkinson’s disease (PD)—a neurodegenerative disorder (NDD) characterized by progressive destruction of dopaminergic neurons within the substantia nigra of the brain—is associated with the formation of Lewy bodies containing mainly α-synuclein. HDL-related proteins such as paraoxonase 1 and apolipoproteins A1, E, D, and J are implicated in NDDs, including PD. Apolipoprotein J (ApoJ, clusterin) is a ubiquitous, multifunctional protein; besides its engagement in lipid transport, it modulates a variety of other processes such as immune system functionality and cellular death signaling. Furthermore, being an extracellular chaperone, ApoJ interacts with proteins associated with NDD pathogenesis (amyloid β, tau, and α-synuclein), thus modulating their properties. In this review, the association of clusterin with PD is delineated, with respect to its putative involvement in the pathological mechanism and its application in PD prognosis/diagnosis.

## 1. Parkinson’s Disease

Parkinson’s disease (PD) is a neurodegenerative disorder (NDD) impairing motor functions; it affects mainly older individuals but also populations below 65 years old (nearly 25% cases) [1,2]. For unclear reasons, PD occurrence has rapidly increased within the last 20 years, doubling in prevalence [2]. The hallmark of PD is a progressive destruction of dopaminergic neurons within the substantia nigra of the brain. The affected nerve cells lose the ability to produce dopamine, which is associated with the formation of Lewy bodies (LBs) containing mainly aggregates of α-synuclein [1,3]. The mechanism of LB generation includes the aggregation process (termed “cross-seeding”) during which α-synuclein binds with other protein complexes including tau protein and β-amyloid yielding intracytoplasmic inclusions consisting of more than 70 proteins [4]. Apart from PD, α-synuclein aggregation is a characteristic feature of other conditions attributed to synucleinopathies, such as dementia with Lewy bodies (DLB), multiple system atrophy (MSA), and pure autonomic failure (PAF) [5]. Except for age, being the main risk factor associated with the development of PD, other agents which increase the odds of PD onset comprise environmental issues such as rural living and herbicide/insecticide exposure, as well as head injuries and sedentary lifestyle leading to obesity [2,6,7]. Additionally, multiple genetic mutations have been implicated in PD pathology [8]; hence, it is believed that there are unfavorable interactions between environmental and genetic risk factors yielding the resultant shape of PD-characteristic neural degeneration [2]. Recently, the greatest meta-analysis of genome-wide association studies (GWASs) has identified as many as 90 independent genetic risk signals, connected mainly with nervous tissue, responsible for 16–36% of heritable PD [9]. Monogenic PD forms occur in a minority and are typical for young-onset disease (below 40 years old) [2].

### 1.1. Faulty Genes and Their Products in the Pathomechanism of PD

Among a variety of PD-associated mutated genes, *SNCA, LRRK2, PRKN, PINK1, DJ-1,* and *GBA* have attracted special attention [2,10]. Genetic implications in parkinsonism pathology with respect to the impact on the impairment of intracellular machinery functioning have been recently profoundly reviewed by Smolders and Van Broeckhoven [11]. Below, the most prominent findings are highlighted.

#### 1.1.1. SNCA

*SNCA* is a gene coding for α-synuclein—a protein present mainly in the presynaptic terminals of nerve cells where it regulates the process of neurotransmitter-transporting vesicle trafficking and budding [12], thus being implicated in the release of dopamine. The most common *SNCA* alterations in PD lead to the improper folding and/or overexpression of its product, which causes the accumulation of toxic α-synuclein aggregates in neurons, resulting in their death. Although the exact mechanism triggering the generation of pathological α-synuclein forms and their toxicity is not known, it includes protein phosphorylation (α-synuclein phosphorylated at Ser129 prevails in LBs) as well as the disruption of plasma membranes via pore generation by toxic α-synuclein oligomers, which further enables calcium influx inducing cell death [4]. Moreover, toxic α-synuclein conformers seem to be able to act in a prion-like mode; they migrate to the adjacent cells, “infecting” their normal counterparts and forcing the change of their conformation from harmless disordered monomers/α-helical oligomers into oligomeric/fibrillar conglomerates containing mostly β-sheet structures [4,12,13]. *SNCA* mutations affect the rate of fibril formation as well as the ability of α-synuclein binding with lipids, impairing the positive functioning of the protein and favoring its toxic conformations (discussed by Meade et al. [12]). Aside from presynaptic terminals, α-synuclein is also present in other intracellular organelles, such as the nucleus, mitochondria, endoplasmic reticulum, Golgi apparatus, and endolysosomal system, where its physiological functions are not fully elucidated; however, pathological forms interfere with the performance of these compartments, and these issues have been recently reviewed by Bernal-Conde et al. [4]. Based on data derived mainly from in vitro cell line or animal studies, pathological α-synuclein has been demonstrated to impair intracellular machinery controlling a variety of functions required for the maintenance of proper homeostasis, including autophagy (chaperone-mediated and/or mitophagy), as well as protein trafficking/turnover machinery (endolysosomal and/or ubiquitin–proteasome system (UPS)) [4]. While properly functioning monomeric α-synuclein regulates some mitochondrial functions (fusion–fission, respiratory chain, and transport systems embedded in the outer mitochondrial membrane such as voltage-dependent anion channels (VDACs)), its toxic forms seem to be responsible for mitochondrial damage and fragmentation. Pathological events stimulated by toxic α-synuclein may be associated with the inhibition of import of cytosol-synthesized proteins being the components of the respiratory chain complexes such as complex I (via blocking the translocase of the outer membrane (TOM)). Additionally, α-synuclein putatively regulates the activities of complexes I and V of the respiratory chain; hence, in a toxic form, it disturbs their functions, which leads to the decrease in ATP synthesis and increase in reactive oxygen species (ROS) production finalized by such cellular responses as mitophagy or apoptosis [4]. Other processes possibly impaired by toxic α-synuclein include gene expression (through the impact on transcriptional regulation by the interaction with epigenetic factors) as well as vesicular trafficking (affecting endoplasmic reticulum, Golgi apparatus, and endolysosomal system) [4].

In a search for putative interactions of α-synuclein with other proteins, Bernal-Conde et al. [4] analyzed STRING and BioGRID databases. They reported the most evident interactions of α-synuclein with protein kinases LRKK2 and SEPT4 (involved in cytoskeleton regulation), PINK1, Fyn, and MAPK1, as well as ubiquitin ligases PARK2, PARK7, and STUB1. Among them are proteins coded by genes whose mutated forms are implicated in PD, such as *PARK2 (=PRKN), PINK1, PARK7 (=DJ-1),* and *LRRK2.*

#### 1.1.2. PINK1 and PRKN

*PINK1* (PTEN-induced kinase 1) and *PRKN* (Parkin RBR E3 ubiquitin-protein ligase) genes encode enzymes involved in the initiation of the mitophagy process, a type of selective autophagy (“self-eating”) aimed at the clearance of dysfunctional mitochondria. Under stressful conditions (such as oxidative imbalance—hypoxia or ROS generation), the potential of mitochondrial membranes is lowered or lost, which prevents *PINK1* product—serine/threonine kinase—from being degraded. PINK1 kinase via phosphorylation processes stimulates the recruitment of the *PRKN* product—E3 ligase Parkin. In turn, active Parkin catalyzes ubiquitination of multiple mitochondrial proteins, which is recognized by ubiquitin-binding cargo receptors. Subsequently, the mitochondria become isolated from the cytoplasm by being surrounded by double membranes; an autophagosome is formed, which further fuses with the lysosome (yielding an autophagolysosome). Finally, lysosomal hydrolases degrade the contents of mitochondrial macromolecules [14]. Rakovic et al. [15] have proposed a modified pathway in the mechanism of PINK1/Parkin-mediated mitophagy, which considers the involvement of UPS before the final degradation of mitochondrial contents by lysosomal hydrolases. Disturbances in damaged mitochondria clearance, caused by faulty PINK1/Parkin signaling, have been associated with a variety of disorders, including NDDs (recently discussed by Quinn et al. [16] and Tanaka et al. [17]). Loss-of-function mutations in *PRKN* and/or *PINK1* being responsible for around 15% (*PRKN*) and up to 4% (*PINK1*) of early-onset parkinsonism implies that the accumulation of damaged mitochondria precipitates the destruction of dopaminergic neurons via the stimulation of oxidative stress and inflammatory processes [11,16,18]. Additionally, considering a regulatory function of Parkin in mitochondrial biogenesis [11], a faulty *PRKN* gene would impair the generation of mitochondria, accelerating pathological events.

#### 1.1.3. PARK7 (=DJ-1)

Loss-of-function mutations in the *PARK7* gene are associated with up to 1% of early-onset PD [11]. Its product DJ-1 is a multifaceted protein controlling the functions of mitochondria, and similarly to PINK1 and Parkin, DJ-1 is involved in the process of mitophagy [11]. Through its cysteine thiol group being oxidized by hydrogen peroxide, DJ-1 detects cellular oxidative stress by which it is activated and triggers protective/antioxidative processes [19]. For example, it regulates the expression of antioxidative enzymes. Moreover, it seems to participate in a chaperone-mediated α-synuclein degradation in lysosomes [20], as well as the synthesis of catecholamines (including dopamine) [21]. Therefore, DJ-1 deficiency is implicated in the disturbance of mitochondrial function, ROS generation, α-synuclein aggregation, and impairment in catecholamine homeostasis.

#### 1.1.4. LRRK2 (=PARK8)

The *LRRK2* gene, whose seven different mutations account for 3–41% of familial PD cases (but are also associated with sporadic types of PD) [2,22], encodes a large multidomain protein termed leucine-rich repeat kinase 2 (LRKK2) [22]. LRKK2 contains both kinase and GTPase activities which are impaired by PD-associated mutations leading to increased kinase activity or decreased GTPase activity [22]. As discussed by Roosen et al. [22], the available evidence suggests the regulatory function of LRKK2 in the processes involved in vesicle trafficking, endocytosis, and autophagy. Pathological LRKK2 forms have been demonstrated to inhibit the process of autophagy by decreasing LC3 lipidation (inevitable for the elongation of a phagophore) but have also been associated with the accumulation of autophagic vesicles, as well as α-synuclein aggregates. Since LRKK2 interacts with and phosphorylates Rab GTPases being involved in the regulation of vesicular dynamics, its defective forms would impair the control over endolysosomal/retrograde/autophagic processes [22,23].

#### 1.1.5. GBA

One of the most commonly mutated genes in PD patients is the beta-glycosylceramidase (*GBA*) gene encoding a lysosomal enzyme, glucocerebrosidase (GCase—D-glucosyl-N-acylsphingosine glucohydrolase), which conducts the hydrolysis of glucocerebrosides (by the removal of glucose moiety). The impairment of the *GBA* gene is known to be responsible for the onset of Gaucher’s disease (GD), which is one of the hereditary lysosomal storage diseases affecting various organs, including the central nervous system (CNS) [24]. However, mutated *GBA* has been also demonstrated to significantly increase PD risk, as recently reviewed in [25,26]. Faulty GCase does not hydrolyze sphingolipids, which tend to accumulate in lysosomes, leading to their impairment. The buildup of GCase substrates (glycosylceramides) and/or fall in its products (ceramides) probably promote and stabilize toxic α-synuclein aggregates and impair α-synuclein secretion [27]. Additionally, both GCase and α-synuclein proteins seem to interact, which is associated with the acceleration of the pathological events; e.g., α-synuclein overexpression disturbs GCase trafficking from the endoplasmic reticulum to the lysosomes, which in turn impairs this enzyme’s hydrolytic function, resulting in glucocerebroside accumulation [28].

## 2. Lipid and Lipoprotein Dyshomeostasis in PD

### 2.1. Intracellular Organelle Impairment Due to Protein/Lipid Dyshomeostasis

In light of the aforementioned genetically conditioned disturbances in cellular homeostasis, it may be imagined that the major PD pathologies are associated with the formation of protein/lipid aggregates, which result from the dysregulation of the machinery involved in the generation of ATP/ROS (mitochondria), molecular trafficking (vesical transport including endo- and exocytosis, as well as lysosomal trafficking), and macromolecular turnover systems (lysosomal-associated autophagy, UPS). A recent study [29], demonstrating the conglomerates of fragmented membranes, vesicles, and organelles (including damaged mitochondria), interspersed with α-synuclein inclusions, in Lewy bodies from PD individuals, supports the hypothesis on the dysfunctional organelles which, together with toxic α-synuclein conformers, are accumulated in dopaminergic neurons, leading to their severe destruction. However, when it comes to the mechanism triggering pathological changes, there are some new concepts focusing on lipid dyshomeostasis being responsible for the generation of toxic proteins/enzymes, which impairs cellular functioning and accelerates pathological events in the mode termed “bidirectional pathogenic loop”. Such an idea has been recently proposed and discussed by Fanning et al. [30,31], who have underlined the growing number of PD-associated faulty genes whose products participate in lipid metabolism and vesicular trafficking, as well as alterations of lipid profile in PD patients’ body fluids. In favor of their hypothesis, the authors have additionally highlighted α-synuclein interactions with lipids and lipid-rich structures, and hence they suggest the potential mutual interplay between altered lipids impairing the functioning of α-synuclein (and other proteins) and toxic α-synuclein conformers destabilizing lipid structures (mainly phospholipid membranes sequestering intracellular organelles).

### 2.2. Lipoprotein Disturbances in PD

Lipid abnormalities associated with inflammatory and oxidative processes in PD (as well as in other NDDs) have been very well documented both at the systemic level in the cardiovascular circulation and in the central nervous system, mainly in cerebrospinal fluid (CSF). For example, in their recent plasma metabolomics analysis, Hu et al. [32] have reported that the majority (over 60%) of differently expressed molecules between PD and control group comprised lipid and lipid-associated molecules (with 25% of sphingolipids), as well as reporting that apolipoproteins B, C, and M are decreased in PD. Decreased levels of ApoB-100 in PD have been also reported in earlier studies [33], whereas other experiments have demonstrated lower levels of ApoE [34], ApoAII [34], and ApoAI [35] in PD patients. On the other hand, data derived from about 600,000 individuals, with over 20-year follow-up, have shown an inverse correlation between the levels of both total and LDL cholesterol, triacylglycerols, and apolipoprotein B and PD risk; a higher level of these lipid markers was significantly indicative of a lower PD risk [36]. Therefore, lipid disturbances must somehow be implicated in PD pathogenesis.

### 2.3. High-Density Lipoproteins (HDLs) and Their Accessory Proteins in NDDs

High-density lipoproteins are (beside VLDL, LDL, and chylomicrons) the means of lipid transport in the cardiovascular system, being involved in cholesterol scavenging from peripheral tissues/blood vessels to the liver (termed “reverse cholesterol transport”) as well as supplying VLDL and chyl with apolipoproteins required for their functioning [37]. Additionally, HDL particles play an antioxidative role (e.g., protecting LDLs from oxidation); hence, they decrease the risk of atherosclerosis and resulting disorders (such as myocardial infarction or stroke). HDL-associated proteins (in the number of around 200 ([38] and references therein)) are necessary for the processes of cholesterol transport (e.g., ABCA1 transporters drawing cholesterol from peripheral cells, and LCAT enzymes esterifying cholesterol which enables packing cholesteryl esters inside HDL particles). Other HDL accessory proteins include apolipoproteins A, C, D, E, F, J, L, and M, as well as paraoxonases 1 and 3 (PON1 and PON3) [38,39]. Due to the diversity of proteins associated physically and functionally with HDL particles, these lipoproteins might be imagined as vehicles of regulatory proteins mechanistically involved in the modulation of inflammatory and hemostatic processes, attenuation of oxidative stress, and neutralization of toxins, rather than mere cholesterol and phospholipid transporters [38,40]. The CNS contains high quantities of lipids such as cholesterol and sphingolipids, which are important components of membranes and are especially responsible for the proper structure of the myelin sheath insulating neuronal axons [41]. Therefore, lipid metabolism and distribution in this tissue are of exceptional significance, and their impairment is implicated in NDD pathologies. When compared with lipoprotein fractions present in blood plasma, only HDL-like particles are found in the CNS, with the major apoproteins being ApoE (synthesized in the CNS) and ApoA-I (delivered from the circulation) [41]. Additionally, these CNS-specific HDL particles contain ApoA-II, ApoCs, ApoD, and ApoJ, as well as using (similar to what occurs in the cardiovascular system) proteins required for cholesterol transport (ABCA/G transporters and LDL receptors/LDL receptor-related protein 1) and esterification (LCAT) [41].

Disturbances in HDL lipid/protein profiles have been implicated in multiple NDDs [42] and were recently reviewed by Marsillach et al. [43] and Cervellati et al. [40], who focused on the implication of apolipoproteins A-I, E, and J (clusterin) [43], as well as PON1 [40,43], in Alzheimer’s disease (AD). Moreover, the involvement of PONs 1–3 in multiple sclerosis (MS), amyotrophic lateral sclerosis (ALS), AD, and PD has been discussed by Reichert et al. [44], and a systematic review on PON1 activity and polymorphisms in MS has been presented by Salari et al. [45]. Additionally, the connection of PON1 with depression, bipolar disorder, generalized anxiety disorder, and schizophrenia has been reviewed by Moreira et al. [46].

In comparison with such NDDs as AD, much less scientific data are available on HDL accessory proteins with respect to PD, and as aforementioned, lipid dysregulation is clearly implicated in PD. Apart from other PD-associated proteins, α-synuclein, the main neurotoxic factor in this disease, has also been implicated in lipid metabolism/transport due to its interactions with lipids and apolipoprotein-characteristic structural features [47].

#### 2.3.1. Apolipoproteins A1, D, and E in PD

Apolipoproteins A1, D, and E are HDL components both in the blood plasma HDL fractions and in HDL-related particles in the CNS.

Apolipoprotein A1 is the major structural component of HDL particles in the circulation. It is mainly synthesized in the liver and intestine (but not in the CNS), and to make part of HDL-like particles in the CNS, it is probably transported to the CNS vehicled by small HDL particles via SR-BI-mediated uptake and transcytosis [48]. Apolipoprotein D is a protein showing homology with the lipocalin family (proteins involved in the transport of small hydrophobic molecules) [49]. Hence, it binds mainly arachidonic acid but also steroid hormones and other lipid compounds [49]. Unlike most proteins, it is poorly expressed in the liver and intestine but widespread in other organs and fluids, such as the spleen, brain, and CSF [49,50]. In the CNS, it is synthesized by astrocytes and oligodendrocytes, as well as neurons and perivascular cells [48,49,50]. It is putatively involved in the protection from inflammatory processes due to its arachidonic acid binding ability (thus restraining arachidonic acid from the conversion into pro-inflammatory molecules such as prostaglandins) [49]. Apolipoprotein E is the major constituent of HDL-related particles in the CNS, where it is synthesized by astrocytes and microglia (but not neurons under normal conditions) [48]. All of these apolipoproteins are implicated in NDDs, probably protecting from neurodegeneration via attenuating oxidative/inflammatory processes associated with lipid transport.

Concluding from many studies, ApoA1 tends to be downregulated in PD patients [34,35,51,52]. Moreover, its decreased levels have been correlated with early PD onset, and the disease progression (reflected by motor functions worsening) [53,54,55]. Also, observed in human populations decreased PD risk in the aftermath of statin treatment, has been ascribed to statin-induced elevated concentration of ApoA1/HDL [56,57]. Additionally, ApoA1 has been proposed as a PD diagnostic marker [55] (showing sensitivity and specificity comparable with α-synuclein) [58]. Generally, ApoA1 seems to play a protective function lowering PD-characteristic neurodegeneration.

Much less is known about ApoD correlation with PD. As discussed by Li et al. [50] it may be involved in the protection from dopaminergic neurons degeneration through capturing arachidonic acid and preventing the generation of proinflammatory eicosanoids. ApoD gene is the most upregulated in the aging brain in many species, including humans, and hence it has been suggested to be involved in the homeostatic mechanism protecting the aging brain from deterioration [59]. The life-extending and ROS-protective properties of ApoD have been demonstrated in *Drosophila* fly models [60,61]. Moreover, the neuroprotective function of ApoD has been corroborated in ApoD-deficient mice whose brains demonstrated increased deterioration associated with many aging characteristics such as the reduction in neuron number and memory deficits [62]. Therefore, ApoD presumably plays a protective function restraining neurodegeneration (via attenuating proinflammatory/pro-oxidative processes). Nevertheless, there are scarcely any observations on the association of ApoD with PD. In their pilot study, Waldner et al. [63] reported that ApoD was increased in the blood plasma of PD patients in comparison with age-matched controls. Additionally, the authors observed a positive correlation between ApoD increase and PD progression. Nevertheless, more research on ApoD in the PD context should be performed to elucidate its function in PD pathology.

ApoE has been thoroughly studied with respect to NDDs, including PD, especially in regard to its polymorphism frequency. There are three alleles of the *APOE* gene (ε2, ε3, and ε4), giving six genotypic variants (E2E2, E2E3, E2E4, E3E3, E3E4, and E4E4) [50]. The most common is the “normal” ε3 allele (reaching 70–80% frequency), whereas ε2 and ε4 are generated through one amino acid substitution in position 112 (ε4) or 158 (ε2) [50,64]. Despite the rich literature data on the implications of ApoE and its isoforms in PD, there is still no consensus on which isoforms, or if any at all, are associated with PD onset and progression. As discussed by Li et al. ([50] and references therein), there are studies demonstrating the correlation of frequencies of both ε2 and ε4 alleles with higher PD risk, whereas in other reports either ε2 or ε4 allele has been more implicated. On the other hand, some studies have shown no association between ApoE isoforms and PD. The soundest scientific evidence substantiates ApoE4 prevalence in PD with cognitive impairment (PDD). However, there are also some inconsistent findings in this field. Some studies have demonstrated that ApoE4 is implicated in the worsening of cognitive functions in PDD [65] and increased risk of PDD [66]. Other scientific reports have shown inconclusive results [67] or no ApoE polymorphism effect on dementia in PD [68,69]. However, most recent scientific data seem to support APOE4 involvement in dementia risk and/or impairment of cognitive functions in PDD [70,71,72]. For example, Davis et al. [73] have corroborated the pathological impact of ε4 in PD. The authors observed the fastest rate of cognitive deterioration in APOE ε4/ε4 patients. Additionally, in mice models expressing different human variants of *APOE* (A53T/E2, E3, E4), they noted the greatest formation of toxic α-synuclein agglomerates associated with gliosis in A53T/E4 mice, whereas ε2 variants exhibited protective function reducing α-synuclein pathologies and increasing the survival of mice, as well as improving their motor performance. Therefore, in light of the recent research, it might be concluded that the ApoE4 variant is associated with the increased risk of PDD and additionally might be involved in the development of toxic conformers of α-synuclein, whereas ApoE2 may act in an opposite way, protecting from neurodegeneration [73].

#### 2.3.2. Paraoxonases in PD

Paraoxonases (PON1–3) comprise a group of three homological enzymatic proteins encoded by adjacent genes located on human chromosome 7, with PON2 regarded as an ancestral one [44,74]. They are widespread in different tissues [75] but synthesized mainly in the liver [44]. PON2 is found only intracellularly, whereas PON1 and PON3 are released into the circulation and are combined with HDL lipoproteins [44,74]. The major HDL-associated paraoxonase is PON1, whereas PON3 contributes only marginally to total PON activity associated with HDL fraction [44]. PON1 demonstrates multiple activities (paraoxonase, lactonase, esterase); therefore, it is implicated in the antioxidative/anti-inflammatory processes [44]. Due to its lipolactonase activity, it protects HDL and LDL lipids from oxidation, and it decreases the oxidative status of macrophages and stimulates cholesterol efflux from macrophages [76], as well as detoxifying a proatherogenic compound—homocysteine thiolactone [77,78]. Additionally, PON1 downregulates monocyte chemoattractant protein (MCP)-1 and myeloperoxidase, the factors involved in vascular inflammation and oxidative stress [77,78]. Considering these anti-inflammatory, antioxidative, and detoxifying functions of PON1, it also seems to be protective against NDDs. In particular, PD onset, whose increased risk is associated with environmental pesticides, should be compromised by high levels of PON1 in the organism, due to the fact that PON1 neutralizes organophosphorus (OP) compounds, such as paraoxon. As discussed by Cervellati et al. [40], experiments on PON1-deficient animals have demonstrated their impaired protection against OP toxicity [79].

Two common single-nucleotide polymorphisms of *PON1* are observed in the human population: Q192R and L55M. The first one affects the ability of PON1 to hydrolyze OP compounds (e.g., paraoxon), whereas the second influences PON1 concentration, with PON1_M55_ being synthesized at a lower level [40,76]. From the studies on human populations exposed to OP molecules, it seems that the PON1_Q192_ alloform gives better protection from toxic compounds than PON1_R192_ [40]. Therefore, a potential association of PON1 activities and its polymorphic variants has been analyzed in PD patients, yielding somewhat contradictory results, probably associated with the specificities of the studied populations. For example, as discussed by Reichert et al. [44], in a Japanese cohort, the PON1_R192_ variant prevailed in PD patients in comparison with the healthy control [80], but in other populations (Caucasian and Chinese) no correlation between PON1 polymorphism and PD has been found [81,82,83,84]. Similarly, the frequency of the M55 allele (producing a decreased concentration of PON1) has been associated with higher PD risk in some studies [85,86] but not in others [83,84]. In meta-analysis studies assessing the level of significance of these two polymorphisms in PD, no impact of Q192R variations on PD status has been reported [87,88], whereas L55M polymorphism was significantly associated with PD in the first study [87], but no correlations were reported in the latter analysis [88]. Therefore, it might be suspected that both polymorphisms are not at all, or only weakly, associated with the disease, and their impact may be dependent on the specificity of the population, as well as the living area. For example, in rural areas where dwellers are exposed to OP compounds (e.g., the components of insecticides), more beneficial PON1 polymorphism associated with faster metabolism of OP molecules would be protective not only from the potential poisoning but also from the development of PD [89,90].

## 3. Clusterin (CLU, ApoJ)

Apolipoprotein J, also termed clusterin, has attracted our attention due to its multifunctionality as well as putative neuroprotective activity exerted at various levels. Therefore, its characteristics with respect to the mechanistic involvement in PD protection/pathogenesis, as well as its evaluation in PD prognosis and diagnosis, are discussed in this and the subsequent sections.

### 3.1. Clusterin Biosynthesis in the Cell—Multiple Products

Clusterin (also termed ApoJ, SPG2, TRPM-2, SP-40,40, and CLI—dependent on the source of its discovery and described characteristics) is a ubiquitous glycoprotein present intra- and extracellularly in many organs (mainly in the CNS, but also in the liver, testis, prostate, pancreas, etc.), as well as in body fluids (including serum, cerebrospinal fluid, mother’s milk, semen, and urine) ([43] and references therein, [91,92]).

The clusterin gene in humans is the size of 18,115 bp and is localized on the short arm of chromosome 8p21-p12 [93]. The gene consists of 11 exons ranging from 126 to 412 bp [94,95]. There are three transcriptional isoforms of human clusterin, which probably have three different transcription initiation start sites [95]. Transcription results in three mRNA isoforms that differ in the fragments being products of exon 1 [94,96]. In the canonical pathway, translation leads to the formation of pre-proprotein consisting of 449 amino acids, where the first 22 constitute the N-terminal signal of translocation to the endoplasmic reticulum (ER) [94]. After the translocation to ER, the pre-proprotein is shortened by the cleavage of a signal peptide to 427 amino acids and is further processed by glycosylation of six Asn residues to produce high-mannose glycoprotein [91]. Additionally, 5–6 disulfide bridges between Cys residues are formed in the ER [97]. At this stage, a 60 kDa precursor, the so-called pre-secreted form of clusterin (psCLU), is formed [98]. The psCLU is translocated to the Golgi apparatus where further glycosylation with galactose, fucose, mannose, N-acetylglucosamine, and N-acetylneuraminic acid takes place, resulting in the formation of an 80 kDa protein that is subsequently cleaved by a furin-like protease between Arg 205 and Ser 206 (Arg 227 and Ser 228 in a pre-proprotein) to form two chains, namely N-terminal α and C-terminal β, and the subunits are joined together by disulfide bridges [98]. Maturation of clusterin involves also sulfation, iodination, and phosphorylation [99]. The mature secreted form of clusterin (sCLU) is a heterodimer composed of two subunits, α and β, with a mass of 40–45 kDa each, joined together by disulfide bridges [98].

Additionally, under stress conditions, noncanonical, cytosolic forms of CLU can be produced. They can arise not only from alternative splicing or alternative start codons during translation, but also from the lack of segregation into ER or retranslocation from ER into the cytoplasm (in such a case they are single-chain proteins since the cleaving into subunits occurs in the Golgi apparatus) [98]. Alternative splicing leads to the production of a truncated form, which does not possess an N-terminal signal of translocation. Therefore, it cannot enter ER and hence lacks the carbohydrate moiety acquired there. This leads to the production of a prenuclear form of clusterin (pnCLU) with a molecular mass of 49 kDa [91]. pnCLU can be converted into a nuclear isoform (nCLU) under stress conditions and translocated to the nucleus where it might be involved in a caspase-3-dependent pathway leading to apoptosis [91].

### 3.2. Clusterin Functions in the Organism

The major heterodimeric secretory form of CLU (sCLU) is one of the most prominent extracellular chaperones [92,98]. This group of proteins includes proteins involved in the clearance of the organism from cellular debris and misfolded proteins, thus protecting from the formation of toxic protein aggregates. sCLU has been demonstrated to form soluble high-molecular-weight (HMW) complexes with such proteins as fibrinogen, albumin, or ceruloplasmin and is seemingly able to remove them from blood plasma in the mode of receptor-mediated endocytosis. In such a way CLU scavenges unwanted proteins from the circulation, delivering the contents for lysosomal degradation in nonprofessional phagocytes (e.g., hepatocytes and endothelial cells [100]) [92,101,102,103,104].

As discussed above, intracellular CLU isoforms are synthesized from truncated mRNA variants (lacking exon 2 nuclear isoform (nCLU)) or derived from immature presecreted forms (psCLU) which evade extracellular trafficking due to post-transcriptional or post-translational modifications [91]. Although their functioning is not fully elucidated, they are believed to regulate cell viability, demonstrating proapoptotic (nCLU) or antiapoptotic (psCLU) properties [91]. However, not all studies have confirmed the apoptosis-modulatory effect of intracellular CLU isoforms [92]. Additionally, the profile of CLU expression seems to be tissue- and cell-specific. For example, the transcript deprived of exon 2, and its product termed nuclear CLU due to its nuclear location observed in some apoptotic cells [105], in astrocytes and neurons is apparently present in the mitochondrial matrix, as demonstrated by Herring et al. [94]. Therefore, the authors hypothesize that in astrocytes and neurons, a specific 45 kD mitochondrial CLU isoform is synthesized, which may regulate the respiratory chain and protect these cells from apoptosis. It might be speculated then, that in CNS this specific CLU isoform takes over the functions ascribed to different intracellular CLU forms in other tissues [106,107,108].

Nevertheless, regardless of the actual isoforms being effective, the major CLU function in the organism seems to be the protection of cells/tissues from different stressors. Clusterin expression is induced in a variety of conditions indicative of cellular/protein dysfunction, such as oxidative stress, and its upregulation is associated with the maintenance of cell viability and survival [109,110]. Therefore, it is considered as a beneficial protein shielding nervous [111] as well as cardiovascular cells [112,113] from pathological events. However, when it comes to neoplastic disorders, it seems to act rather in favor of cancer cells, increasing their metastatic potential [114]. Generally, CLU probably protects cells in the initial stages of injury, but when pathological processes progress and cross a critical point, its capacity becomes insufficient for protection, and hence it even might enhance toxic changes, as is observed in malignant cells [110,111]. Nevertheless, CLU induction upon such stressors as ROS increase has been well substantiated, as discussed by Trougakos [110]. The assumptions supporting this hypothesis are as follows: (1) CLU upregulation commonly observed in diseases associated with oxidative stress such as cardiovascular, neoplastic, and neurodegenerative dysfunctions; (2) CLU upregulation by ROS in cell lines; (3) stress-induced regulatory sequences in *CLU* gene promoter [109,110]. An increase in CLU expression at both mRNA and protein levels has been observed upon induction by oxidants in human fibroblasts [115] and neuroblastoma cells [116]. Additionally, the *CLU* gene promoter contains a few regulatory elements which are responsive to (among others) oxidative stress, such as AP-1 and “CLU-specific element”. The first element is recognized by activator protein-1, and the second by heat shock factor-1 (HSF-1), both of which are mobilized in stress conditions, including oxidative stress [110]. Moreover, *CLU* gene expression is regulated by other signaling molecules associated with inflammatory/oxidative processes, such as NF-κB, or growth factors and cytokines [110]. There are also some studies demonstrating the protective function of CLU against oxidative stress. For example, human sCLU overexpression in *Drosophila melanogaster* protected the fruit flies from oxidative stress, lowered ROS levels in their bodies, and increased their longevity [117]. For these reasons, as proposed by Trougakos and Gonos [109,110], clusterin might be regarded as a biosensor of cellular/oxidative stress, which upon induction triggers the safety mechanisms.

### 3.3. Clusterin in the CNS—Upregulation and Possible Neuroprotection in NDDs

In the CNS, clusterin is synthesized mainly in astrocytes (but not in microglial cells), as well as in neurons. It seems to function intracellularly, protecting neurons from apoptosis and regulating oxidative processes in mitochondria, and extracellularly as a chaperone binding and removing toxic conformers of NDD-associated proteins such as β-amyloid (Aβ), tau protein, and α-synuclein [118,119,120]. Besides, coupled with HDL-like particles, CLU is probably involved in lipid (mainly cholesterol) transport and distribution between glial and nerve cells. The involvement of CLU in NDDs, especially in AD, has been recently addressed in some review papers [43,111,121]. Its upregulation has been observed in dementia (mostly associated with AD [122]), ALS [123], MS [124], TSE [125], and Huntington’s disease [126]. Moreover, positive correlations have been demonstrated between CLU levels and neurodegeneration/atrophy [127,128]. CLU is probably involved in Aβ binding, which prevents the formation of toxic fibrillar forms [119], and enhances Aβ efflux from the brain across BBB via the LRP1/2 receptors [129,130]. Therefore, the role of CLU in AD is assumed as neuroprotective, stabilizing cell viability and inhibiting Aβ accumulation/toxicity (together with ApoE [131]) [132,133,134] and/or enhancing Aβ clearance [132,134,135], as well as improving cognitive functions (at least in the Tg6799 mice model [132]). Although there are some data suggesting proamyloidogenic actions of CLU in AD [136,137], those are in the minority, and although they support the effects of CLU on Aβ dynamics, they seem to reach contradictory conclusions. For example, the first experiments of DeMattos et al. [137] have suggested that CLU promotes Aβ toxicity, but in their subsequent findings [131] the collaboration of CLU with ApoE in limiting the generation of toxic Aβ aggregates (and resulting neurodegeneration) has been observed. Nevertheless, although CLU seems to function mainly towards neuroprotection, there are studies that suggest its neurotoxic properties, as observed in the recent experiments by Yuste-Checa et al. [138].

## 4. Clusterin in PD Pathology

Clusterin acts mostly as an extracellular, ATP-independent chaperone. It has been found at the sites of misfolded protein deposit localization [139,140,141]. Further evidence for clusterin involvement in the clearance of aberrantly folded proteins comes from the fact that its expression increases in neurodegenerative diseases, where a toxic effect of such protein aggregates has been documented [142]. In line with this observation, Lenzi et al. [139] demonstrated that clusterin transcription and translation were stimulated along with the increase in α-synuclein concentration. The direct evidence for the engagement of clusterin in protein aggregate clearance comes from the experiment conducted on a rat model by Wyatt et al. [101], who demonstrated more efficient removal of misfolded proteins from circulation when they formed complexes with clusterin.

Aggregation-prone proteins have specific intrinsically disordered domains that facilitate their aggregation [143]. Aggregation of α-synuclein oligomers is accompanied by conformational changes leading to the formation of more compact and stable complexes. Such changes in structure are vital for the subsequent fibrillation process and formation of toxic deposits that are less predisposed to degradation [144]. Clusterin is considered a “holdase”, which means that it can prevent aggregation and precipitation of protein de-posits but is not able to refold misfolded proteins [143]. In this respect, Yerbury et al. [120] demonstrated that clusterin can inhibit the aggregation process of various proteins, among them α-synuclein. Whiten et al. [144] reported that binding between clusterin and α-synuclein involves regions of exposed hydrophobicity on the surface of a client protein. They also noted that binding between a given client protein and a chaperone is specific for each chaperone and proceeds through hydrophobic interactions; binding has been inhibited by the application of bisANS, a probe of solvent-exposed hydrophobicity. Each chaperone contains a specific domain necessary to execute binding of a client protein, and in the case of clusterin, it is considered that such domain, functionally similar to α-crystallin, is localized between residues 286 and 343 [145]. Subsequently, Bailey et al. [146] used sequence analysis to reveal that clusterin contains most likely three disordered/molten globule-like domains required for client-protein binding, two of them localized at C- and N-terminal ends of the protein. The binding domains may be buried in aggregates formed by clusterin molecules themselves. In fact, it has been implied that clusterin oligomers may serve as reservoirs of this chaperone [147]. It is suggested that the shift towards lower pH causes dissociation of clusterin molecules from such aggregates. The reduction of pH to mildly acidic shifts the equilibrium between aggregated and heterodimeric forms towards the latter [147]. Poon et al. [147] demonstrated that the changes in pH do not alter the secondary or tertiary structure of clusterin but result in the exposure of hydrophobic regions to the environment. It can be compared to the activation of small heat shock proteins (sHSPs) by temperature. In the case of sHSPs, the higher temperature causes the increase in the rate of subunit exchange in their heterodimers [148,149]. In the case of clusterin, temperature has no effect on the aggregation state, but pH seems to induce similar changes [147]. As a result, hydrophobic regions of clusterin are exposed to the environment and the chaperone can more effectively bind client proteins [150,151]. The experiments of Poon et al. [147] revealed that the dissociation of aggregates at mildly acidic pH is most probably due to the protonation of histidine residues in clusterin structures. Protonation of histidine residues may lead to the disruption of interactions between clusterin heterodimers in the clusterin aggregates. Subsequently, heterodimers are released with the hydrophobic regions ready to bind to the hydrophobic domains of client proteins [147]. In fact, the chaperone activity is enhanced at lower pH and therefore can more efficiently work extracellularly at locally lower pH values [147,151]. What is more, local acidosis can be detected at sites of tissue damage or inflammation [152,153]. Such a phenomenon has been also detected in the brains of Alzheimer’s patients [154]. Therefore, it seems that the shift of clusterin assemblies towards its heterodimer units may be an example of physiological adjustment to the conditions which may, otherwise, lead to the progression towards pathology [150].

The exact way by which protein aggregates exert their toxic effects is not known, but it is implied that by binding to cellular membranes they can change their permeability, which leads to ROS production. This hypothesis is supported by the experiments conducted by Whiten et al. [144], who found that binding of α-synuclein by clusterin lowered lipid membrane permeability and reduced ROS generation.

The binding of misfolded proteins is only an initial step in the prevention of protein deposit cytotoxicity. The second role of a chaperone is to facilitate their neutralization or degradation. In the case of extracellular clusterin, after binding with misfolded protein aggregates, the complexes can directly bind to the cell surface receptors, followed by internalization and subsequent intracellular proteasomal or lysosomal degradation [101,155]. In fact, it has been demonstrated that clusterin augments proteasome function by binding to ubiquitin [156]. It has also been documented that clusterin may be involved in plasminogen activation system clearance of toxic protein deposits through the formation of complexes with plasmin-generated protein fragments, with subsequent internalization via receptors and intracellular degradation [157]. Clusterin–client protein complexes can be internalized by cell surface receptors belonging to the low-density lipoprotein receptor family, such as megalin, the very low density lipoprotein receptor, and ApoER2, as well as plexin [158,159].

Although clusterin predominantly functions in the extracellular space, it can also act inside the cells, as has been already described, especially during ER stress. During ER stress, clusterin is retained in the cytoplasm, where it can bind to the aberrantly folded proteins and traffic them for degradation either in proteasomes or via autophagy ([158] and references therein). 

It has been estimated that clusterin binds to a client protein at clusterin–client protein ratio between 1:0.33 and 1:5 [98]. It is interesting that depending on the ratio between clusterin and client protein, the function and the outcome of the chaperone action can change. If this ratio is high, as has been described on the example of interactions between clusterin and Aβ, then clusterin exerts its neuroprotective function, but when this ratio is very low, which means the Aβ is at a very large molar excess, it becomes even cytotoxic and induces oxidative stress [120]. Li et al. [159] suggested that at a low concentration of the chaperone, clusterin can bind to hydrophobic regions of a client protein but cannot prevent its cytotoxicity. Chaplot et al. [143] on the other hand proposed that when the concentration of client protein is very high, clusterin can act in a sequestering rather than scavenging mode, surrounding the aberrant proteins and potentially facilitating the formation of insoluble protein deposits, making them less toxic. Hence, it is possible that the chaperone action at very high concentrations of an aberrant protein may differ depending on the client protein type and the degree of such excess.

Therefore, it might be speculated that depending on the PD pathology stage, clusterin impact might be initially beneficial, but in the more advanced steps (associated with the robust accumulation of toxic α-synuclein agglomerates) the effects of clusterin may even stimulate neurodegeneration. Such a pathology-favoring phenomenon has been observed by Filippini et al. [141]. The authors reported that clusterin bound and limited the uptake of α-synuclein by human and murine astrocytes.

Hence, the following chain of events may be suggested: In the discrete initial pathological changes triggered by oxidative/inflammatory processes observed during neurodegeneration, clusterin would be induced by ER stress. Then, its intracellular forms would be synthesized simultaneously with extracellular sCLU. Subsequently, clusterin may bind and compromise α-synuclein spreading both intra- and extracellularly. Thus, in neurons, CLU would hold α-synuclein in harmless complexes preventing its toxic interactions with membranes and restraining it from leaving the cells. In the next stages associated with higher levels of toxic α-synuclein being released from neurons, extracellular sCLU (produced by astrocytes and neurons) would surround α-synuclein agglomerates and remove them from extracellular space by binding with respective receptors on phagocyting glial cells (mainly astrocytes). After endocytosis, clusterin/α-synuclein complexes (or only α-synuclein) would undergo proteolysis in UPS and/or lysosomal systems. In the subsequent stages, when α-synuclein would exceed the proportions favorable for clusterin binding, CLU would only surround and restrain the toxic client proteins in the extracellular matrix, in such a way preventing them from spreading and destroying adjacent neurons and glial cells. However, after crossing a kind of a critical point (such as an accumulation of an excessive amount of fibrillary toxic α-synuclein conformers), clusterin, being still induced by oxidative/inflammatory stress, would itself reach toxic levels, which at this late stage would only contribute to neurodegeneration. These events would run in parallel with the degeneration of astrocytes not capable of further hydrolysis of uptaken toxic aggregates. Nevertheless, since the above scenario is merely a hypothesis, this issue requires further investigation with respect to neurodegenerative effects of aberrant folding of various proteins and their interactions with clusterin, also in the aspect of clinical application of such information.

The potential modes of action of clusterin, in regard to its extracellular chaperone function, are presented in Figure 1.

## 5. Diagnostic Potential of Clusterin in PD

In light of the predicted doubling of PD prevalence over the next 30 years, there is a growing need for a reliable disease risk assessment, identification in earlier—prodromal—phase, and improving the accuracy of disease diagnosis [160]. Like in other NDDs, the PD diagnosis is based on clinical presentation, currently supported by neuroimaging, but can only be definitely confirmed postmortem. The PD diagnosis remains challenging due to the substantial symptom overlap with other NDDs, particularly with the movement disorders referred to as atypical parkinsonian disorders. Among those are dementia with Lewy bodies (DLB), multiple system atrophy (MSA), corticobasal degeneration (CBD), and progressive supranuclear palsy (PSP) [160]. Misclassification is quite common, and the accuracy of clinical criteria-based diagnosis is less than 80% [161]. Moreover, mixed NDD pathology frequently revealed upon autopsy may compound clinical diagnosis and influence the biomarker profile in cerebrospinal fluid (CSF) [162].

The onset of cardinal motor PD features corresponds with substantial, up to 80%, loss of dopaminergic neurons (reviewed in [163]). It may be preceded by other, nonspecific symptoms (prodromal phase), such as REM sleep behavior disorder (RBD); hyposmia; depression; and autonomic dysfunction manifested as constipation, urinary urgency, erectile dysfunction, or orthostatic hypotension. RBD in particular seems to be a valuable clinical marker of PD risk as up to 90% of patients will eventually develop clinically defined PD or PD with dementia (PDD) [160,164].

The CSF is a well-established source of biomarkers for central nervous system diseases, including NDDs, as their concentration in other body fluids is usually extremely low. However, it is only accessible through invasive procedures; therefore, noninvasive or minimally invasive systemic biomarkers are sought after, particularly for screening and disease monitoring purposes [163,165]. Although the nonclinical PD biomarkers of the prodromal phase are needed to allow for developing early disease-modifying therapies [160,164], both CSF and systemic biomarkers for PD have received relatively little attention. Moreover, the available studies have produced rather unimpressive and/or inconsistent results [163,166]. They are frequently conducted on small cohorts of living patients and thus without a definite confirmation by a postmortem analysis of both the disease pathology and the absence of other abnormalities, potentially affecting clusterin concentration. Additionally, the varying degree of contamination of analyzed CSF samples with erythrocytes, a well-known clusterin source, is likely to contribute to the reported discrepancies between studies. 

The diagnostic potential of clusterin in the differential diagnosis of neurological diseases was first suggested by Polihronis et al. [167]. The authors observed that NDDs and meningitis are not associated with significantly altered CSF clusterin, contrary to demyelinating diseases and acute neuropathologies, such as cord compression or epilepsy. In those, clusterin concentration was significantly higher than in the reference group, consisting of patients undergoing spinal anesthesia and individuals with headache or back pain. Clinical studies regarding clusterin in PD are scanty and inconclusive, with the bulk of research focusing on cerebrospinal fluid (CSF)-derived protein and its usefulness in differential diagnosis.

### 5.1. Clusterin in PD Diagnosis

#### 5.1.1. Cerebrospinal Fluid (CSF) Clusterin

The CSF clusterin is suggested to originate from the brain and not the blood [168]. However, erythrocyte presence may falsely elevate clusterin level in the CSF [169] since the protein is relatively abundant in extra- and intracellular membranes as well as the cytoplasm of red blood cells [170].

Comparing CSF clusterin in AD, vascular dementia, and PD has corroborated findings on unaltered protein level in NDDs [167] by showing the lack of significant between-group differences, as well as no difference as compared to healthy individuals [171]. Subsequently, comparable CSF clusterin concentration between neurologically healthy individuals and PD patients was reported in a larger cohort by van Dijk et al. [169]. In addition, the authors did not observe any correlation between CSF clusterin and patients’ gender or age or the disease duration, stage (determined using Hoehn and Yahr classification, or severity expressed using the motor subscale of the Unified Parkinson’s Disease Rating Scale (UPDRS-III). Moreover, CSF clusterin did not differ between distinct clinical PD subtypes, namely between younger-onset PD, tremor-dominant PD, and non-tremor-dominant PD, and displayed no correlation with cognitive decline expressed in terms of Mini-Mental State Examination (MMSE) score.

Contrary results were first presented by Yin et al. [172], who reported a 1.5-fold elevation of clusterin in pooled CSF from three PD patients as compared to three healthy controls, determined using one-dimensional electrophoresis followed by liquid chromatography–tandem mass spectrometry (LC-MS/MS). The slight elevation has also been observed in two-dimensional electrophoresis for multiple clusterin spots, representing various post-translationally modified isoforms, ranging from 1.2- to 1.4-fold upregulation in PD. Likewise, clusterin has been found to be elevated in AD, up to 1.6-fold. Subsequent studies by Přikrylová Vranová et al. [173,174,175] corroborated clusterin elevation in PD using quantitative ELISA. They have also observed an inverse relationship with the disease duration as patients with the onset of the disease below two years had significantly higher CSF clusterin than those with PD for two years or more [173]. This observation has raised the possibility of clusterin being an early PD marker. However, it was not corroborated by the aforementioned larger study of van Dijk et al. [169].

CSF clusterin has been found to be elevated not only in PD but also in PDD, while it has not differed from controls in AD, DLB, MSA, or PSP [174,175]. Since PD and PDD patients had significantly higher CSF clusterin than patients with atypical parkinsonian disorders (DLB, MSA, and PSP), the potential of this protein as a differential marker has been evaluated using receiver operating characteristic (ROC) curve analysis. Clusterin differentiated PD from atypical parkinsonian disorders with 84% accuracy and displayed 83% sensitivity and 76% specificity at 5.76 µg/mL cut-off. Subgroup analysis has shown clusterin to be the best marker differentiating PD from MSA, as it displayed 91% accuracy and sensitivity and specificity of 91% and 82%, respectively, at 4.5 µg/mL cut-off. As a diagnostic marker distinguishing PD from neurologically healthy individuals, clusterin had an accuracy of 80% and sensitivity and specificity of 83% and 71%, respectively, at 5.65 µg/mL cut-off [175]. However, because the number of patients in each group was small and nonoptimal for ROC analysis, the obtained characteristics of clusterin as a diagnostic and differential marker in PD should be interpreted with caution, especially in light of the fact that a larger study has not confirmed clusterin elevation at all [169], and all three studies by Přikrylová Vranová et al. [173,174,175] seem to have highly overlapping cohorts and should rather be treated and weighted as one. The discrepancies between studies can also be attributed to the higher erythrocyte count in the PD cohort in the study of Přikrylová Vranová et al. [173] compared to that in the study of van Dijk et al. [169]. It has been recommended that CSF samples with a red blood cell count above 500 per microliter should not be used for biomarker studies [176], and the maximal erythrocyte count in the cohort of Přikrylová Vranová et al. [173] was 1194 per microliter with median 27 per microliter in PD and 7 per microliter in PDD cohorts, as opposed to 1 per microliter in their own reference group or the PD cohort in the study of van Dijk et al. [169].

Relatively high misclassification rate and uncertainty of pure PD diagnosis with the potential impact of other copathologies on clusterin concentration may also contribute to the lack of consistency between studies. Maarouf et al. [177] were the only authors to conduct postmortem clusterin analysis, showing the protein increased in ventricular CSF of neuropathologically confirmed PD patients, as compared to the reference material, by 2-fold. Pooled PD and, separately, control samples have been analyzed using two-dimensional difference gel electrophoresis (2D-DIGE) with protein identification by MALDI-TOF/TOF/MS.

A summary of the studies focused on CSF clusterin in PD is presented in Table 1.

#### 5.1.2. Systemic Clusterin

Two-dimensional liquid chromatography–tandem mass spectrometry (2D-LC-MS/MS) coupled with isobaric tags for relative and absolute quantification (iTRAQ) labeling was used by Zhang et al. [35] to demonstrate higher clusterin concentration in serum of PD patients as compared to controls, with more pronounced elevation in stages I–II (by 2.8-fold) than III–IV (by 1.4-fold) according to Hoehn and Yahr classification. These observations contradict the findings of van Dijk et al. [169] as the authors consistently reported no difference between PD patients and healthy controls and no association with the disease duration, stage, severity, or clinical subtype or MMSE score in plasma clusterin.

Apart from serum and plasma, clusterin level has recently been determined in exosomes released by neuronal cells and present in circulation [178,179,180]. These small (30–120 nm) endosome-derived vesicles are believed to reflect the proteome of cells from which they originate and are considered an attractive source of biomarkers as they can be captured in living patients with minimal invasiveness. Kitamura et al. [178] compared exosomal clusterin between PD patients and healthy controls with reference to the disease progression. The 2D-DIGE and MALDI-TOF/TOF/MS analysis revealed three spots corresponding to clusterin isoforms. Exosomal clusterin was significantly decreased in PD patients with Hoehn and Yahr stages II (by 1.5- to 1.8-fold) and III (by 1.8- to 2.0-fold) as compared to healthy individuals but was not listed among plasma proteins significantly different between PD patients and healthy controls. Jiang et al. [179,180] applied multiplexed electrochemiluminescence to determine clusterin in serum exosomes immunocaptured with neural cell adhesion molecule L1 (L1CAM). They demonstrated an elevation in clusterin in frontotemporal dementia (FTD), PSP, and CBD—NDDs with dominant tau or TAR DNA-binding protein (TDP)-43 proteinopathy but minimal α-synuclein pathology—as compared to healthy individuals. In turn, exosomal clusterin in PD, PDD, DLB, MSA, and RBD was comparable between these patients and was not significantly changed as compared to healthy controls. However, patients with clinical PD (PD and PDD) or prodromal PD (represented by RBD) had lower exosomal clusterin than those with alternative proteinopathies. The protein accuracy was 79% as a differential clinical PD marker and 83% as a differential prodromal PD marker. However, in combination with α-synuclein, respective accuracies were improved, reaching 98% for differentiation of both clinical and prodromal PD from other proteinopathies. These excellent accuracies were accompanied by high sensitivities (94% and 95% for clinical and prodromal PD, respectively) and specificities (96% and 93% for clinical and prodromal PD, respectively) [179]. In their follow-up study [180], the authors used the same methodology to validate α-synuclein-to-clusterin ratio as a stratification tool in an enlarged population of patients. Previous observations on elevated exosomal clusterin in tau-related atypical parkinsonian syndromes have been confirmed. The calculated α-synuclein-to-clusterin ratio (α-Syn/Clu×1000) had improved overall accuracy, consistently in both training and validation cohorts (98 and 99%, respectively), over α-synuclein alone in discriminating PD from PSP and CBD. Sensitivities (93% and 100% for training and validation cohorts, respectively) and specificities (98 and 95% training and validation cohorts, respectively), corresponding with α-Syn/Clu×1000 > 1.1 cut-off, were superior as well.

A summary of the studies focused on systemic clusterin in PD is presented in Table 2.

### 5.2. Clusterin in PD Risk Assessment

Clusterin gene (*CLU*) has been identified as a susceptibility genetic locus for a late-onset AD by genome-wide association studies (GWASs) and a meta-analysis [181], prompting research on the potential utility of *CLU* as a prediction marker for other NDDs, including PD [182,183]. Gao et al. [182] analyzed the distribution of rs11136000, a single-nucleotide polymorphism in the *CLU* gene, among 791 PD patients (of which 19% had dementia) and 1580 matched controls, reporting that carriers of both T alleles (TT as compared to CT and CC) had a lower PD risk with odds ratio (OR) = 0.71 (95% CI: 0.55–0.92). The risk was greater for PDD (OR = 0.49, 95% CI: 0.27–0.91) than PD without dementia (OR = 0.81, 95% CI: 0.61–1.06). In a subsequent longitudinal study, Sampedro et al. [183] demonstrated that high-risk naïve PD patients, defined as carriers of C allele (CC or CT), were more likely to develop mild cognitive impairment (PD with MCI) or dementia (PDD), with relative risk ratio (RR) = 1.91 (95% CI: 1.1–3.4). They also performed worse on various cognitive assessments, including the Symbol Digit Modality Test (SDMT), Hopkins Verbal Learning Test (HVLT), Benton Judgment of Line Orientation (BLJO), and Montreal Cognitive Assessment (MoCA) at baseline and/or at the 5-year follow-up.

## 6. Concluding Remarks

Judging from the general functions of clusterin in the organism, mainly with respect to its putative protective role as a stress-induced extracellular chaperone, it might be hypothesized that in the CNS it protects nerve and glial cells from such stressors as ROS or inflammatory processes, and especially misfolded proteins. Such proteins as Aβ, tau, or α-synuclein, involved in the generation of pathological events in neurodegenerative diseases such as AD and PD, have been demonstrated to be neutralized by clusterin. Therefore, it might be suggested that clusterin is involved in scavenging toxic proteinous aggregates. However, since there are also reports suggesting its activities in the direction of cytotoxic aggregate promotion, it might be hypothesized that the result of clusterin action (harmful or beneficial) depends on the intensity or degree of the observed pathology. Additionally, since a variety of faulty genes and environmental factors are implicated in NDDs such as PD, different pathological routes lead to characteristics for PD degeneration, which makes it difficult to assess the impact of one chosen factor. On the other hand, regardless of the exact mechanism, the common feature accompanying PD-characteristic dysfunctions is oxidative stress. Therefore, oxidative-stress-induced clusterin would rather work in favor of nerve and glial cells, triggering mechanisms aimed at oxidative stress reduction. Furthermore, in light of clusterin multifunctionality, it might be speculated that different, still unknown, signaling pathways resulting in cell protection may be implicated in its actions. Nevertheless, more research is required to elucidate the multifaceted function of clusterin in the human organism under normal conditions, as well as its role during the development of a variety of pathologies, including PD. A comparable level of complexity is seen in the estimation of the potential of clusterin in the diagnosis and prognosis of PD. Firstly, there are very few studies assessing the potential of clusterin in diagnosis and prognosis. Secondly, a high level of uncertainty in the diagnosis of “pure” Parkinson’s disease makes it difficult to draw unambiguous conclusions and contributes to the observed inconsistencies between studies.

## Figures and Tables

**Figure 1 antioxidants-11-00524-f001:**
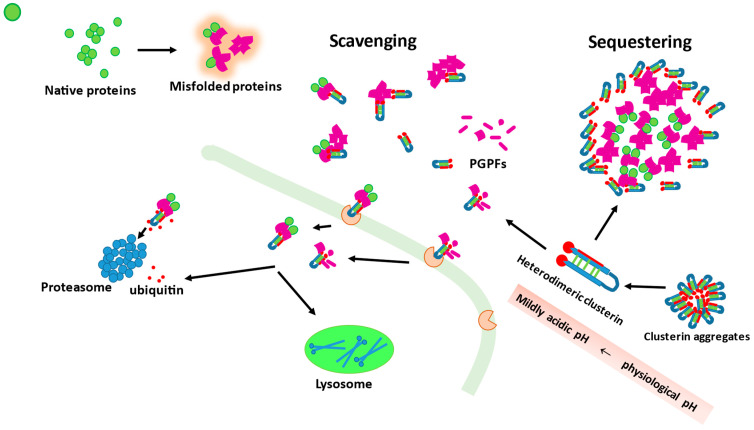
Clusterin molecules can form aggregates that serve as their reservoirs. Clusterin can act as a scavenging or sequestering agent. As a scavenger agent, it can bind misfolded, toxic proteins as well as plasmin-generated protein fragments (PGPFs). The complexes between misfolded protein (or PGPFs) and clusterin are recognized by membrane receptors and trafficked inside the cell where they are degraded either by proteasomes or lysosomes. In an excess of client protein molecules, clusterin can work as a sequestering agent surrounding misfolded toxic molecules and thus preventing their toxicity. Dissociation of clusterin aggregates into heterodimeric fully active forms is enhanced when pH changes from physiological values to mildly acidic.

**Table 1 antioxidants-11-00524-t001:** Summary of studies on cerebrospinal fluid (CSF) clusterin in Parkinson’s disease.

Reference Group		PD		Other Conditions		Observations	Methodology		Ref.
*N*	C (µg/mL)	*N*	C (µg/mL)	*N*	C (µg/mL)				
NHC (11)	5.6 ± 1.1	18	3.4 ± 1.1	AD (32)	6.5 ± 1.9	no btw-group differences	LP	ELISA ^2^	[171]
				VAD (20)	6.6 ± 1.6				
NHC ^1^ (50)	4.6 ± 1.1	52	4.9 ± 1.4	-	-	no btw-group differences; no effect of PD duration, stage (H&Y), or severity (UPDRS-III); no differences btw clinical PD subgroups ^3^	LP	ELISA ^2^	[169]
NHC (3)	-	3	-	AD (3)	-	1.5-fold ↑ in PD but not AD	LP	1-DE MS	[172]
						Multiple spots: up to 1.4-fold ↑ in PD and 1.6-fold in AD		2-DE MS	
NHC ^4^ (30)	4.7	32	9.0	-	-	↑ in PD as compared to NHC; ↑ in PD < 2 yrs. as compared to ≥2 yrs.	LP	ELISA ^5^	[173]
NHC ^4^ (24)	4.5 (2.3–9.3)	27	6.3 (4.1–14)	PDD (14)	8.9 (2.3–12.1)	↑ in PD as compared to NHC; ↑ in PDD as compared to NHC (tendency: *p* = 0.052); otherwise no significant btw-group differences	LP	ELISA ^5^	[174]
				DLB (14)	4.9 (1.9–8.4)				
				AD (17)	7.3 (3.0–16.4)				
NHC ^4^ (21)	4.4	23	6.9 (4.1–14)	PDD (18)	8.6 (4.0–13.0)	↑ in PD as compared to NHC, DLB, PSP, MSA; ↑ in PDD as compared to NHC, DLB, MSA; AUC ^6^: PD vs. NHC—80%, PD vs. DLB—82%, PD vs. MSA—91%, PD vs. PSP—82%, PD vs. DLB/MSA/PSP—84%	LP	ELISA ^5^	[175]
				DLB (15)	4.2 (1.9–8.5)				
				MSA (12)	4.2 (1.3–6.9)				
				PSP (16)	3.6 (3.1–9.2)				
				AD (18)	5.2 (1.4–10.2)				
NHC (49)	-	43 ^7^	-	-	-	↑ by 2-fold in PD as compared to NHC	V-CSF	2D-DIGE/MALDI	[176]

↑, clusterin elevation; 2D-DIGE/MALDI, two-dimensional difference gel electrophoresis (2D-DIGE) with protein identification by MALDI-TOF/TOF/MS; AD, Alzheimer’s disease; btw, between; C, concentration; D, group characteristics; DE MS, one- or two-dimensional electrophoresis followed by mass spectrometry analysis; DLB, dementia with Lewy bodies; ELISA, enzyme-linked immunosorbent assay; H&Y, Hoehn and Yahr classification; LP, cerebrospinal fluid collected by lumbar puncture; MSA, multiple system atrophy; *N*, number of observations; NHC, neurologically healthy controls; PD, Parkinson’s disease; PDD, PD with dementia; PSP, progressive supranuclear palsy; Ref., reference; UPDRS-III, the motor subscale of the Unified Parkinson’s Disease Rating Scale; VAD, vascular dementia; V-CSF, ventricular CSF obtained postmortem; yrs., years. ^1^, self-declared healthy volunteers; ^2^, in-house test; ^3^, younger-onset/tremor-dominant/non-tremor-dominant; ^4^, patients with vertebrogenic or psychogenic disease, migraine, tension headache, or diabetic neuropathy; ^5^, commercial test (Biovendor, Czech Republic); ^6^, diagnostic accuracy determined using receiver operating characteristic (ROC) curve analysis and expressed as area under the curve (AUC) in percent; ^7^, neuropathologically confirmed PD diagnosis.

**Table 2 antioxidants-11-00524-t002:** Summary of studies on systemic clusterin in Parkinson’s disease.

Reference Group		PD		Other Conditions	Observations	Methodology		Ref.
*N*	C	*N*	C	Disease (*n*): C (ng/mL)				
NHC (6) ^1^	-	16 (8 + 8) ^1,2^	-	-	↑ in H&Y I–II PD as compared to NHC by 2.8-fold; ↑ in H&Y III–IV PD as compared to NHC by 1.4-fold	S	iTRAQ/2D-LC-MS/MS	[35]
NHC ^3^ (50)	76.1 ± 12.6 µg/mL	52	73.2 ± 11.1 µg/mL	-	no btw-group differences; no effect of PD duration, stage (H&Y), or severity (UPDRS-III); no differences btw clinical PD subgroups ^4^	P	ELISA ^5^	[169]
NHC (8) ^1^	-	16 (8 + 8) ^1,6^	-	-	↓ in H&Y II PD as compared to NHC by 1.5- to 1.8-fold; ↓ in H&Y III PD as compared to NHC by 1.8- to 2.0-fold; no significant difference btw H&Y PD II and III	E	2D-DIGE/MALDI	[178]
NHC (144)	8.7 ± 4.9 ng/mL	275 ^7^ (230 + 45)	9.7 ± 6.0 ng/mL	DLB (21): ns ^8^	no btw-group differences for PD, PDD, DLB, MSA, RBD, and NHC; ↓ in PD as compared to PSP, FTD, and CBD; ↓ in RBD, DLB, and NHC as compared to PSP, FTD, and CBD; αSyn/Clu ↑ in PD, PDD, and RBD as compared to NHC; αSyn/Clu ↑ in PD, PDD, RBD, and DLB as compared to PSP, FTD, and CBD; AUC ^9^: PD/PDD vs. other PPs—79% for Clu and 98% for Clu and α-Syn; RBD vs. other PPs—83% for Clu and 98% for Clu and α-Syn	E	ECL	[179]
				MSA (14): 6.8 ± 3.2				
				PSP (35): 18.4 ± 8.8				
				CBD (45): 16.2 ± 6.1				
				FTD (65): 20.2 ± 10.5				
				RBD (65): 10.0 ± 5.2				
NHC ^10^ (191/47)	13.0 ± 5.2 ng/mL	290/60 ^10^	12.1 ± 4.8 ng/mL	MSA (50/36): 12.2 ± 7.0	↑ in PSP and CBD as compared to NHC, PD, and MSA; AUC ^9^ for αSyn/Clu×1000 (training and validation cohorts): PD vs. NHC—78% and 89%; PD vs. MSA—97% and 89%; PD vs. PSP/CBD—98% and 99%	E	ECL	[180]
				PSP (116/81): 20.0 ± 7.2				
				CBD (88/43): 21.2 ± 9.6				

↑, clusterin elevation; 2D-DIGE/MALDI, two-dimensional difference gel electrophoresis (2D-DIGE) with protein identification by MALDI-TOF/TOF/MS; αSyn/Clu, α-synuclein-to-clusterin ratio; btw, between; C, concentration; CBD, corticobasal degeneration; Clu, clusterin; D, group characteristics; DLB, dementia with Lewy bodies; E, exosomes; ECL, multiplexed electrochemiluminescence; FTD, frontotemporal dementia; H&Y, Hoehn and Yahr classification (PD stage); MS, mass spectrometry; MSA, multiple system atrophy; *N*, number of observations; NHC, neurologically healthy controls; ns, not specified; PD, Parkinson’s disease; PDD, PD with dementia; PPs, proteinopathies; PSP, progressive supranuclear palsy; RBD, rapid eye movement (REM) sleep behavior disorder; Ref., reference; S, serum; UPDRS-III, the motor subscale of the Unified Parkinson’s Disease Rating Scale. ^1^, analyzed on pooled samples; ^2^, PD patients stratified based on PD stage into H&Y I–II and III–IV with eight patients in each group; ^3^, self-declared healthy volunteers; ^4^, younger-onset/tremor-dominant/non-tremor-dominant; ^5^, in-house test; ^6^, PD patients stratified based on PD stage into H&Y II and III with eight patients in each group; ^7^, endosomal clusterin concentration has been given for combined PD (*n* = 230) and PDD (*n* = 45) patients; ^8^, 7.0 ± 3 and 6.2 ± 5.2 ng/mL (separate data for two cohorts); ^9^, diagnostic accuracy determined using receiver operating characteristic (ROC) curve analysis and expressed as area under the curve (AUC) in percent; ^10^, a follow-up study on enlarged cohort—both total number of participants and new participants are indicated and separated by a slash symbol.

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
