# Peer review of "HDL Accessory Proteins in Parkinson’s Disease—Focusing on Clusterin (Apolipoprotein J) in Regard to Its Involvement in Pathology and Diagnostics—A Review"

_antioxidants, 2022, doi:10.3390/antiox11030524_

Round 1

Reviewer 1 Report

This paper is a review that focuses on clusterin (apoJ) and its potential role in the pathology of Parkinson’s disease.  The authors start by an overview of PD pathology, followed by a section on 5 genes and their products that are involved in the physiopathology of PD.  The next section focuses on disturbances of lipoprotein metabolism in this disease, particularly HDL and their accessory proteins. Section 3 revies the role of clusterin, in both systemic and CNS, it’s potential role(s) in PD, concluding with the potential of clusterin as diagnostic biomarker or in risk assessment of PD, ending with general concluding remarks.

General comments

I found this review to be generally well-written and very thorough, providing, for the reader, sufficient detail and pertinent information on the results/conclusions obtained by the different articles cited.  The manuscript is well organized with regards to the different topics addressed.   The subject is timely and provides up to date information on clusterin in not only PD but neurodegenerative disorders in general. 

Major comments

Line 283 –“ Clusterin expression is induced in a variety of conditions indicative of cellular/protein 283 dysfunction, such as oxidative stress”. In view of the target journal, could the authors provide some additional information on the specific role of clusterin in oxidative stress?  What results obtained support this conclusion?  Were there any indication in diagnostic or risk assessment of association of clusterin protein/gene with those involved in oxidative stress and ROS production?  A statement or two regarding clusterin and oxidative stress in the conclusions would also be helpful. 

Line 426_427. Regarding studies on clusterin as biomarkers, can the authors provide more details on the inconsistent results, and speculate why this is the case?

Minor comments:

Title – Since the authors use primarily the term “clusterin” in the body of the manuscript, apolipoprotein J should be in parentheses rather than clusterin.

Section 1.2 “Intracellular organelles impairment due to protein/lipid dyshomeostasis. This does not seem to belong to Section 1 on the PD genes.  This seems more appropriate in Section 2, which could be retitled to something such as “Lipid and lipoprotein dyshomeostasis in PD”. 

Figure 1, It would be interesting to indicate (schematically) how this could protect from oxidative stress. line 398 – Delete the last two sentences “This is a figure…formatting”.

Tables 1 and 2 – The list of numerous abbreviations are very helpful, please list in alphabetical order, this would be easier for the reader.   

Lines 533 and 537 – should this be exosome rather than endosome since you are discussing detection in individuals? Please clarify.    

Word usage –

 Some sentences are rather long, and could be split into two sentences (for example, lines 549-552). 

Although the word pathomechanism is used the literature, for me this is imprecise and not in accordance with English word usage. Please use another term such as “pathological mechanism(s)” or “physiopathology”.

All “et al” should be in italics

Line 499 – “dedicated” should be replaced with “focused on”

List of abbreviations – This is greatly appreciated and useful, please also add “PDD”.  Also, “Alzheimer’s disease” should be AD in the text after the first usage – please verify (lines 443, 461, 576).

Author Response

Answer to Reviewer 1

On behalf of all authors, I would like to thank you for your valuable comments which hopefully have improved our manuscript. Below, please find the detailed answers addressing your remarks.

Major comments

Line 283 –“ Clusterin expression is induced in a variety of conditions indicative of cellular/protein 283 dysfunction, such as oxidative stress”. In view of the target journal, could the authors provide some additional information on the specific role of clusterin in oxidative stress?  What results obtained support this conclusion?  Were there any indication in diagnostic or risk assessment of association of clusterin protein/gene with those involved in oxidative stress and ROS production?  A statement or two regarding clusterin and oxidative stress in the conclusions would also be helpful.

Answer

Thank you for this comment.

The more detailed information on clusterin in oxidative stress has been added at the end of section 3.2. (Clusterin functions in the organism), as well as in section 6 (Concluding remarks).

When it comes to the significance of clusterin in diagnosis and prognosis of oxidative stress-associated diseases, there are literature data assessing its importance in regard to specific diseases such as cardiovascular, neurodegenerative, as well as neoplastic disorders. All of these disorders comprise a heterogenous groups of multifactorial diseases/syndromes. Regardless of their separate pathological backgrounds, they have one common feature associated with their pathological mechanisms, which is oxidative stress. Nevertheless, the studies evaluating disease-applicable markers come from material derived from patients suffering from defined disorders, and in our work we focused on Parkinson’s disease (PD), since there have been recent reviews describing clusterin utility as a marker in other diseases such as Alzheimer’s disease. Therefore, we decided to accumulate and discuss the experimental reports concerning mostly PD, since there is comparably less information in the literature databases in the aspect of clusterin utility in PD diagnosis/prognosis.

Line 426_427. Regarding studies on clusterin as biomarkers, can the authors provide more details on the inconsistent results, and speculate why this is the case?

Answer:

Thank you for this comment. The indicated statement has been extended and the following phrase was added: “They are frequently conducted on small cohorts of living patients and thus without a definite confirmation by a post-mortem analysis of both the disease pathology and the absence of other abnormalities, potentially affecting clusterin concentration. Additionally, varying degree of contamination of analyzed CSF samples with erythrocytes, a well-known clusterin source, is likely to contribute to the reported discrepancies between studies.”

Please, note that a more detailed analysis of between-studies discrepancies and their source is conducted in section 5.1.1., in which the contradictory results are presented and discussed.

Minor comments:

Title – Since the authors use primarily the term “clusterin” in the body of the manuscript, apolipoprotein J should be in parentheses rather than clusterin.

Answer:

Thank you for this remark which has been addressed; the terms in the Title have been exchanged

Section 1.2 “Intracellular organelles impairment due to protein/lipid dyshomeostasis. This does not seem to belong to Section 1 on the PD genes.  This seems more appropriate in Section 2, which could be retitled to something such as “Lipid and lipoprotein dyshomeostasis in PD”.

Answer:

This section was meant to be kind of a connection between sections 1 and 2, and actually it looks better as an opening of section 2. Therefore, the proposed rearrangements have been introduced.

Figure 1, It would be interesting to indicate (schematically) how this could protect from oxidative stress. line 398 – Delete the last two sentences “This is a figure…formatting”.

Answer:

The sentences have been removed. However, clusterin protection from oxidative stress, despite its well substantiated connection, is not elucidated at the mechanistic level. Therefore, regretfully, we have not been able to propose a logical scheme, which would illustrate this issue. Instead, we added the effect of pH reduction on the mobilization of active clusterin dimers from its inactive aggregates.

Tables 1 and 2 – The list of numerous abbreviations are very helpful, please list in alphabetical order, this would be easier for the reader.

Answer:

Thank you for this suggestion. The abbreviations in Table 1 and 2 footnotes have been listed in an alphabetical order.

Lines 533 and 537 – should this be exosome rather than endosome since you are discussing detection in individuals? Please clarify.

Answer:

Thank you for pointing it out. Endosomes have been changed into exosomes in both indicated cases.

Word usage –

Some sentences are rather long, and could be split into two sentences (for example, lines 549-552).

Answer:

Thank you for this suggestion. The indicated sentence as well as several other long sentences have been rewritten and shortened. 

Although the word pathomechanism is used the literature, for me this is imprecise and not in accordance with English word usage. Please use another term such as “pathological mechanism(s)” or “physiopathology”.

Answer:

The word “pathomechanism” has been replaced by other terms (e.g. in Title, as well as in section 4 this word has been substituted by “pathology”)

All “et al” should be in italics

Answer: The formatting has been corrected

Line 499 – “dedicated” should be replaced with “focused on”

Answer: It has been replaced

List of abbreviations – This is greatly appreciated and useful, please also add “PDD”.

Answer: This acronym has been added to the Abbreviations; PDD - Parkinson's disease with dementia

Also, “Alzheimer’s disease” should be AD in the text after the first usage – please verify (lines 443, 461, 576).

Answer:  “Alzheimer’s disease” has been replaced by AD in the indicated lines.

I hope you will find the revised manuscript acceptable for publication.

With best regards,

Izabela Berdowska

Reviewer 2 Report

The review about the potential role of clusterin in Parkinson`s disease by Berdowska and colleagues is interesting, and such, fills a gap in the field. The review is well-written, however, some parts can be improved.

As the review focuses on Clusterin, and to a less extend, to HDL proteins, the lengthy part about PD-related genes/proteins is not necessary. There is plently of literature for that, and it is not easy to summarize the whole field in such short text. 

Rather, the part with HDL proteins should contain more detail and with focus more on PD. A more descriptive introduction to HDL-related proteins, with functions and tissue distribution would be very useful. E.g. the authors mention PON1, but gives almost no information about its functions, only refer the reviews for several lines.

Regarding clusterin, I miss a more precise description of the structure/synthesis of the protein. The first paragraph is hard to follow - eg it is not clear that the α and β subunits are cleaved intracellularly from the precoursor, and they are forming in antiparallel location the complex. It is also somehow hidden in the text, that the majority of the protein is synthetised from one major transcript, all the othrs are minor forms. The origin of the intracellular protein is not exactly clear. 

Similarly, a better description of the pH-dependent chaperone activity would be useful. 

It would also be interesting to mention, that it is missing from microglia. 

As alpha-syuclein pathology occurs predominantly intracellularly, a more clear statement or lengthier speculation is missing, how exactly the authors suggest, that clusterin may protect neurons or increase toxic effect of synuclein? As the paragraph is about PD, probably at least synucleinopathy-specific facts would fit better here. 

A speculation might be for protective effect as the reduction of intercellular spreading of aSyn pathology; the better clearance of tissue by phagocytotic cells. However, one should cite contradictory inications as well: eg Filippini at all found that clusterin might be responsible for decreased clearance of aSyn fibers from the extracelleular space - therefore might facilitate propagation

As for diagnostic: probably a schematic diagram or table would help the reader to digest the many numbers and facts – summarizing the findings from CSF, serum, plasma, endosomes, etc

Finally, please carefully revise the English/grammar – many sentences are hard to read/understand.

.

Author Response

Answer to Reviewer 2

On behalf of all authors, I would like to thank you for your time spent on the review of our manuscript. We are grateful for the valuable remarks and hope that they have contributed to the improvement of the submitted paper. Below, please find the detailed answers to your comments.

Reviewer’s comment 1:

As the review focuses on Clusterin, and to a less extend, to HDL proteins, the lengthy part about PD-related genes/proteins is not necessary. There is plently of literature for that, and it is not easy to summarize the whole field in such short text.

Answer:

Thank you for your remark.

Parkinson’s disease (PD) is a common neurodegenerative disorder with still unclear multifactorial pathological background, hence actually a lot of literature is found on this topic in scientific databases. However, we thought that an introductory part focused on the most impactful genetic failures associated with the disturbances of important players involved in the regulation of cellular organelle functioning, would be beneficial for “Antioxidants” journal readers who not necessarily are experts in neurodegenerative diseases (including PD). Additionally, we have tried to underline the implications of the impaired genes’ products in the cellular machinery disturbances. Some of them affect mitochondrial functioning, others lipid homeostasis. Moreover, most of these dysfunctions are associated with oxidative stress. Similar disturbances are discussed in the subsequent sections with reference to PD and lipid dyshomeostasis, as well as clusterin function in this disease. For example clusterin in a way resembles a PD-associated gene (PARK7) product – DJ-1 which detects the level of oxidative stress, triggers antioxidative mechanisms, as well as plays the function of a chaperone protein. To make our work more comprehensible, we have introduced more explanation associated with the antioxidative significance of clusterin at the end of section 3.1. (Clusterin forms and functions in the organism), as well as in section 6 (Concluding remarks).

 We fully agree that the multiple genes involved in the pathological mechanism of PD make this issue really thorough and complicated, but we have tried to show this complexity and heterogeneity of PD in a concise way, at the same time referring the reader to the latest literature for more details. Therefore, we would rather not remove this part from our manuscript.

Reviewer’s comment 2:

Rather, the part with HDL proteins should contain more detail and with focus more on PD. A more descriptive introduction to HDL-related proteins, with functions and tissue distribution would be very useful. E.g. the authors mention PON1, but gives almost no information about its functions, only refer the reviews for several lines.

Answer:

Thank you for this suggestion. Actually, more information on other HDL-associated proteins in PD has complemented our manuscript with interesting issues.

Section 2.3 (High density lipoproteins (HDLs) and their accessory proteins in NDDs) has been enriched in two subsections describing other apoproteins and paraoxonases with respect to Parkinson’s disease (2.3.1. Apolipoproteins A1, D and E in PD; 2.3.2. Paraoxonases in PD)

Reviewer’s comment 3:

Regarding clusterin, I miss a more precise description of the structure/synthesis of the protein. The first paragraph is hard to follow - eg it is not clear that the α and β subunits are cleaved intracellularly from the precoursor, and they are forming in antiparallel location the complex. It is also somehow hidden in the text, that the majority of the protein is synthetised from one major transcript, all the othrs are minor forms. The origin of the intracellular protein is not exactly clear. 

Answer:

The details concerning clusterin biosynthesis have been added at the beginning of Chapter 3. Hopefully, now this part is more informative.

Reviewer’s comment 3:

Similarly, a better description of the pH-dependent chaperone activity would be useful. 

Answer:

Chapter 4 has been enriched in details associated with the proposed mechanism of pH effect on clusterin activation. Moreover, Figure 1 has been completed with a schematic illustration of pH effect on clusterin.

Reviewer’s comment 3:

It would also be interesting to mention, that it is missing from microglia. 

Answer:

This information has been added at the beginning of Chapter 3.3 (“In CNS clusterin is synthesized mainly in astrocytes (however not in microglial cells), as well as in neurons.)

Reviewer’s comments 3 and 4:

As alpha-syuclein pathology occurs predominantly intracellularly, a more clear statement or lengthier speculation is missing, how exactly the authors suggest, that clusterin may protect neurons or increase toxic effect of synuclein? As the paragraph is about PD, probably at least synucleinopathy-specific facts would fit better here. 

A speculation might be for protective effect as the reduction of intercellular spreading of aSyn pathology; the better clearance of tissue by phagocytotic cells. However, one should cite contradictory inications as well: eg Filippini at all found that clusterin might be responsible for decreased clearance of aSyn fibers from the extracelleular space - therefore might facilitate propagation

Answer:

Thank you for these comments. We hope that the presented now speculations have made our work more interesting.

A hypothetical mechanism of clusterin action with respect to the protection from α-synuclein toxicity both intracellularly and extracellulary in the central nervous system has been proposed at the end of Chapter 4. Additionally, the mechanim of potential clusterin toxicity with reference to the indicated Fillipini study has been addressed there.

Reviewer’s comment 5:

As for diagnostic: probably a schematic diagram or table would help the reader to digest the many numbers and facts – summarizing the findings from CSF, serum, plasma, endosomes, etc

 Answer:

Thank you for this suggestion. The findings from CSF and from serum, plasma and endosomes are discussed in details in sections 5.1.1. and 5.1.2 and additionally summarized in two tables (Table 1 and Table 2). One table is dedicated to CSF clusterin and the other to clusterin in systemic circulation. There is no point in presenting serum/plasma and exosomes in separate tables as there is only one study on plasma and one on serum clusterin. We think that these tables are comprehensive summary of available studies – providing information not only on the results but also on their credibility by presenting the number of analyzed cases as well as the technique of clusterin assassement.

The problem with available results is that they are mostly contradictory – e.g. while serum clusterin is reportedly elevated in PD, a study on serum culsterin shows no disease-related changes. Confirmatory studies, if present, are mostly conducted by the same group of authors and analyze the same cohorts of PD patients. As such, there is no clear picture of diagnostic utility of clusterin to be summarized in some simple diagram.

Reviewer’s comment 5:

Finally, please carefully revise the English/grammar – many sentences are hard to read/understand.

Answer:

Thank you for this remark. English language has been checked and corrected. Many too long sentences have been shortened to improve the clarity of the contents.

I hope you will find the revised manuscript acceptable for publication.

With best regards,

Izabela Berdowska